# Effects of Group Emotion and Moral Belief on Pro-Environmental Behavior: The Mediating Role of Psychological Clustering

**DOI:** 10.3390/ijerph191811190

**Published:** 2022-09-06

**Authors:** Shanshan Li, Feiyu Chen, Xiao Gu

**Affiliations:** 1School of Economics and Management, Anhui University of Science and Technology, Huainan 232001, China; 2School of Economics and Management, China University of Mining and Technology, Xuzhou 221116, China

**Keywords:** group emotion, moral belief, pro-environmental behavior, psychological cluster

## Abstract

An important topic in ecological environmental protection concerns the determination of a method to guide individual pro-environmental behavior from the dual perspectives of immediate intervention and long-term shaping. This research analyzed the influence of group emotions and moral beliefs on the pro-environmental behavior of residents and introduces the concept of the “psychological cluster” to explore the mediating effect of the “psychological cluster” on group emotions, moral beliefs, the pro-environmental behavior of residents, and its various dimensions. The results of a questionnaire survey, factor analysis, regression analysis, and other methods showed that group emotions and moral beliefs can significantly the predict pro-environmental behaviors of residents, and moral beliefs have a significant impact on basic pro-environmental behaviors. Each dimension of the psychological cluster has a significant effect on pro-environmental behavior. Among them, the psychological cluster has the most significant effect on the citizen-type pro-environmental behavior. That is to say, when individuals are passionate about environmental causes, and feel angry and guilty about damaging the environment, they will be encouraged to actively participate in environmental protection activities in public places and give advice on environmental protection. Group emotion and moral belief can drive the occurrence of a psychological cluster, which leads to the generation of group behavior. In other words, psychological clustering acts as a “middleman” between group emotion, moral belief, and pro-environmental behavior of residents. However, its influence is not completely mediated via the psychological cluster. In summary, the psychological cluster plays a partial mediating role in the relationship between group emotion, moral belief, and the pro-environmental behavior of residents. Finally, corresponding policy suggestions are put forward based on this research.

## 1. Introduction

Ecological environmental protection is an important topic of common concern worldwide. The interaction between residents and the environment has an important impact on environmental protection [1]. Researchers define individual behaviors that are beneficial to environmental protection and resource conservation as environmentally friendly behaviors, that is, pro-environmental behavior [2,3]. Therefore, it is important to explore the formation mechanism driving pro-environmental behavior in order to guide the emergence of pro-environmental behavior. Many studies have been carried out on pro-environmental behavior from different perspectives, especially related studies on its influencing factors [4,5,6,7]. However, the long-term mechanism and rapid emergence of resident interventions with regard to pro-environmental behaviors still need to be explored further.

The emergence of pro-environment behavior is directly related to psychological factors [8]. At present, the research on the psychological factors affecting pro-environmental behavior mainly focuses on cognitive and affective aspects, such as environmental knowledge, environmental concern, anger, and guilt [9]. Among them, emotion can have an important impact on individuals’ behavior at work [10]. In a group, the emotions between individuals can easily produce a “transmission” effect [11]. In addition, individuals can experience emotions at the group level, including anger, fear, guilt, and pride. Emotions can further regulate the behaviors of individuals and groups [12], and they also affect the specific pro-environmental behaviors of individuals [13].

The influence of group emotion on pro-environmental behavior is transient and fluctuating. Compared to emotion, belief is a steady-state characteristic shown by an individual and has the characteristics of permanence and non-fluctuation [14]. Moral belief is a belief that something is right or wrong—moral or immoral [15]. Research has found that moral beliefs have a certain driving effect on prosocial behaviors, such as donation and volunteering [16]. As a kind of prosocial behavior, moral beliefs may have a long-term and stable promotional effect on pro-environmental behavior.

In the field of social psychology, a cluster, also known as a group, refers to two or more individuals who come together to achieve a specific goal and form an interactive and interdependent relationship. Cluster behavior refers to the action that group members participate in to improve the group’s status quo [17]. Traditional studies on cluster behavior mainly have two orientations, emotion-focused and problem-focused, and emotion-focused refers to the subjective sense of injustice or relative deprivation caused by social comparison [18]. People choose to participate in cluster behavior in order to express the negative emotions caused by a sense of injustice (e.g., anger) [19]. Emotions describe the state of psychological preparation prior to the occurrence of group behavior [20]. Furthermore, in a cluster, individual members usually show group psychological characteristics inconsistent with individual psychological characteristics. Therefore, the group members have similar psychological characteristics [21]; psychological cluster formations, which will affect the real individual psychological characteristics (such as emotion, faith, etc.); and environmental behavioral relationships. In addition, the cluster effect will promote the rapid emergence of pro-environmental behaviors in the residents.

In conclusion, this study aims to discuss the functional mechanisms of group emotions and moral beliefs of residents’ pro-environmental behavior from the perspective of short-term and long-term dual pathway mechanisms. Furthermore, combined with a psychological cluster effect analysis of residents, this study systematically explores the conduction effect of group emotions and moral beliefs on the pro-environmental behavior shown by residents in order to provide relevant references for the promotion of the pro-environmental behavior of residents.

## 2. Theoretical Basis and Research Hypotheses

### 2.1. Pro-Environmental Behavior and Its Structural Dimensions

Pro-environmental behavior, also known as environmentally friendly behavior, reflects the positive attention shown by individuals to environmental issues and their resulting positive attitudes and behavioral tendencies.

Studies on pro-environmental behavior began earlier in countries outside China and are more comprehensive. Early researchers associated pro-environmental behavior with a sense of responsibility. Hines et al. (1987) proposed that behavior concerning solving environmental problems or protecting the environment should be called pro-environmental behavior and that it is influenced by personal values and responsibility [22]. Hsu and Roth (1998) pointed out that pro-environmental behavior appears when people assume responsibility for the social and ecological environment [23]. In recent years, researchers have tended to study and define pro-environmental behavior from the point of view of externality. Udall (2020) refers to behavior that makes beneficial changes to the environment or reduces adverse effects on nature as pro-environmental behavior [24]. Lee et al. (2014) asserted that pro-environmental behaviors are individual behaviors that have positive spillover effects on the ecological environment [25]. Khashe et al. (2015) held the belief that that a pro-environmental behavior means the behavior shown by people that can promote the sustainable development of the environment or reduce the adverse impact of individuals on the environment [26]. Thus, the above-mentioned researchers have defined pro-environmental behavior from different angles.

Research into pro-environmental behavior in China started relatively late. Wu et al. (2006) held that pro-environmental behavior should be defined as behaviors in which individuals take initiative to protect the environment and positively affect it “from the viewpoint of positive influence” [27]. Wang (2015) and other researchers defined the behavior of individuals reducing negative impacts on the environment as pro-environmental behavior from the angle of a negative impact [28].

Although researchers around the globe have slightly different perspectives on the definition of pro-environmental behavior, their connotations are basically the same. Based on the above definitions, the pro-environmental behaviors shown by residents are defined in this study as behaviors of individuals that have positive impacts on the environment or reduce negative impacts on the environment. A structural dimensional study by Chen et al. (2017) on the pro-environmental behaviors shown by individuals suggested that these behaviors can be divided into the following four dimensions: the basic pro-environmental behavior of a resident in public places, such as turning off lights, saving electricity, and garbage sorting; decision-making pro-environmental behavior driven by decision factors at work, such as purchasing environmentally protective products and green office supplies; interpersonal pro-environmental behavior at work due to the influence of interpersonal relationships and social influence; and active participation by residents in social environmental protection issues in public places and offering opinions on environmental protection, which is also called citizen pro-environmental behavior [29].

### 2.2. Research on the Correlation between Group Emotion and Pro-Environmental Behavior

Group emotion originally evolved from the “social emotion” proposed by Smith [30]. The definition of and research into group emotion are based on two different perspectives. Viewed from the object that stimulates the emotion, Smith believed that group emotion is the personal emotional experience of a group or group members. Other researchers, working from the perspective of the subject of the emotions, defined group emotions as the emotions felt by individuals who classify themselves as members of a group [31]. This study, focusing on the subject of emotion, suggested that group emotion is the emotion produced by an individual towards the environmental behaviors and resulting outcomes of its group members, including anger, guilt, anxiety, etc., where the individual ascribes himself/herself into a certain group, such as that of human beings.

There have been few studies about the influence of group emotion on pro-environmental behavior. A comprehensive analysis of the existing literature on group emotion demonstrated that the influence of group emotion, especially group anger, on group behavior is a research topic/subject mostly discussed and studied by scholars. Shearer et al. (2020) found that group emotion is significantly positively correlated with collective efficacy and collective action [32]. Ferguson and Branscombe (2010) in their study concerning the relationship between global warming cognition and environmental behavior reported that group guilt, as a mediating variable, plays a positive role in promoting individual environmental behavior [33]. The influence of group emotion on environmental behavior has received some attention in recent years. Ferguson and Branscombe (2010) put forward that group guilt stimulates its members’ resource-saving behavior. [33]. Group emotion played a mediating role between environmental responsibility and pro-environmental behavioral tendencies in their study about the influence of group emotion on behavioral intentional tendency in the context of air pollution. It is evident that group emotion exerts a certain influence on pro-environmental behavior, but its influencing mechanism remains to be explored. Therefore, this study aims to investigate the influence of group emotion on the pro-environmental behavior shown by residents and proposes the following hypothesis:

**Hypothesis** **1** **(H1).**
*Group emotion has a significant effect on the pro-environmental behavior shown by residents.*


### 2.3. Research on the Correlation between Moral Beliefs and Pro-Environmental Behavior

Moral belief is a strong and absolute belief that something is right or wrong, i.e., moral or immoral [15]. When individuals hold certain moral beliefs, they will think that their position is objectively correct and will feel angry in the face of “immorality” [34]. Based on the definition by Skitka (2002) and the scale developed by Skitka et al. (2005) [35], Maarten et al. (2011) [34], and Wang et al. (2015) [28], this study defined moral belief as a strong and absolute belief that environmental protection is right or wrong—moral or immoral.

The existing studies related to the impact of moral beliefs on behavior are mostly concentrated on social participation, clustering, and consumer behavior. Martijn et al. (2011) claimed that moral beliefs can predict the formation of clusters; that is, the stronger the moral beliefs an individual holds, the higher the probability of said individual participating in the clusters [36]. Maarten (2011) suggested that moral beliefs can promote the generation/occurrence of pro-environmental behaviors to a certain degree, such as donating and volunteering [34]. As a kind of prosocial behavior, moral beliefs also play a certain role in promoting pro-environmental behavior [37]. However, further in-depth study on their predictive effect and influencing mechanism is needed. Hence, this study hypothesized that:

**Hypothesis** **2** **(H2).**
*Moral beliefs have a significant effect on the pro-environmental behaviors shown by residents.*


### 2.4. Research on the Correlation between Psychological Clusters and Pro-Environmental Behavior

Although there is currently no clear definition of a psychological cluster, relevant studies on clusters, group psychology, and psychological belonging have laid the foundation for research into psychological clusters. In sociology and psychology, a cluster, also known as a group, refers to two or more individuals who come together to achieve a specific goal and form an interactive and interdependent relationship. Gustave (2016) believed that a group is “a group of aggregated people”, emphasizing that clustering was not necessarily physical but spiritual [38]. Park (1969) defined a cluster as a group of people gathered under a common focus of concern [39]. Psychological belonging is closely related to groups. Hidalgo (2002) suggested that the sense of belonging is an internal connection between an individual and the group to which they belong [40]. The identification and maintenance of an individual toward the group and its affiliation refer to the degree to which an individual or a group identifies with an event or a phenomenon. Fritsche (2017) suggested in his research on the social identity model of pro-environmental behavior (SIMPEA) that when facing environmental problems, individual actors will transform psychologically into collective actors owing to social identity, that is, the “psychological cluster” mentioned in this study [41]. In summary, a psychological cluster is defined in this study as a phenomenon wherein individuals are gathered psychologically, then transform themselves into collective actors, and thereby generate a psychological sense of belonging.

Based on previous research on clusters, group psychology, and psychological belonging, this study divides psychological clusters into the following three dimensions: (1) a responsibility cluster composed of individuals who believe that they are responsible for environmental protection and thus choose to assume their responsibilities of protecting the environment and participate in the cluster; (2) an emotional cluster comprising individuals who love the cause of environmental protection, feel angry and guilty about any environmentally damaging practice, and then tend to take part in an environmental protection cluster; and (3) a resonance cluster containing individuals who associate the quality of the environment with their own life and survival and then participate in an environmental cluster based on resonance.

Researchers studying the influence of social identity on pro-environmental behavior point out that when confronting environmental issues, individual actors will psychologically transform into collective actors based on social identity [41], that is, they form a “psychological cluster”. The influence mechanism driving psychological clusters’ effects on the pro-environmental behavior of residents needs to be further explored. Therefore, the following hypothesis was formed:

**Hypothesis** **3** **(H3).**
*Psychological clusters have a significant effect on the pro-environmental behaviors shown by residents.*


### 2.5. Research on the Mediating Effect of Psychological Clusters

In the field of organizational behavior and social psychology, psychological variables are often studied as mediating variables. Shi (2010) suggested that individuals are susceptible to herd mentality, which leads to emotional contagion [42]. Based on the theory of group emotion, Li et al. (2018) found that the group behavioral type is an important factor affecting the relationship between group anger and group behavioral intention [43]. Based on the research results of scholars, we find that group emotions and moral beliefs have a certain influence on the formation of psychological clusters.

The psychological cluster also has a certain effect on behavior. Based on the group pressure theory, clustering can lead to the generation of expected behaviors [44]. Liu et al. (2020) found that the sense of belonging can significantly improve the value-co-creative behaviors of community members [45]. Kyei-Poku (2020) undertook a study on the path of leadership influencing organizational citizenship behavior in which he found that belonging has a positive impact on organizational citizenship behavior [46]. Consequently, psychological clusters play a certain role in the relationship between group emotions, moral beliefs, and behavior. Therefore, this study outs forth the following hypotheses:

**Hypothesis** **4** **(H4).**
*Group emotions indirectly affect the pro-environmental behaviors shown by residents through psychological clusters and their various dimensions;*


**Hypothesis** **5** **(H5).**
*Moral beliefs indirectly affect the pro-environmental behaviors shown by residents through psychological clusters and their various dimensions.*


The internal influencing mechanisms are will be clarified so as to effectively guide the residents’ pro-environmental behaviors and stimulate the emergence of such behaviors. From the perspective of psychological clusters, this study mainly focuses on the long-term and short-term dual pathway mechanisms. Based on a literature review, a comprehensive research model was built, as shown in Figure 1.

## 3. Research Design

### 3.1. Variable Measurement

There is a difference between the subject of the emotion and the object of the emotion. The subject of emotion is the individual who can experience the emotion through evaluating an event in line with his/her own goals. The object of emotion refers to some kind of stimuli by which the emotion can be induced. This study, concentrating on the subject of emotion, defined group emotion as the emotion felt by an individual towards the environmental behaviors and resulting outcomes of its group members, including anger, guilt, anxiety and so on, where the individual identifies himself/herself with a certain group (such as human beings). Therefore, this study applied the scale developed by Zomeren et al. (2004) to investigate the influence of group anger, guilt, and anxiety on pro-environmental behavior. On the basis of the literature review above, the definition given by Skitka (2002), and the scale developed by Skitka et al. (2005) and Maarten et al. (2011), moral belief in this study is defined as a strong and absolute belief that environmental protection is right or wrong, i.e., moral or immoral. At present, there is no clear definition of a psychological cluster, but research on clusters, group psychology, and the psychological sense of belonging forms the foundation of research into psychological clusters. Based on the existing literature, this study defines the psychological cluster as a phenomenon through which people are gathered together psychologically, and then transform themselves into collective actors and thus generate a psychological sense of belonging. On this basis, by referring to the existing mature scales devised by Abrahamse and Steg (2009) [47], this study’s initial scale was developed independently. There have been many studies on pro-environmental behavior scales. Although originating from various perspectives, to some extent, these scales still have something in common. In addition, their influencing factors were obtained through literature analyses, and their connotation and measurement have the support of mature scales. Given all this, this study, borrowing the existing mature scales of residents’ pro-environmental behavior and its various influencing factors, and making some improvements and revisions according to experts’ suggestions, built its own brand-new and initial scale. This study used the Likert 5-point method measure, where values from 1–5 mean “strongly disagree” to “strongly agree”, and the higher the score, the higher the degree of recognition by respondents of a certain point of view.

### 3.2. Data Collection, Sample Selection, and Descriptive Statistics

In order to test the feasibility of the scale, a preliminary survey was carried out from 9 April 2021, to 18 April 2021. The stratified sampling method was adopted prior to the pre-investigation so that the rationality of the sample distribution was ensured. The pre-investigation questionnaires were distributed through network channels, and the questionnaire links were sent through WeChat, QQ, Weibo and other social platforms. Moreover, the purpose of the survey was explained in detail to ensure the efficient collection of the questionnaire. A total of 132 questionnaires were sent out and 101 valid questionnaires were received with a recovery rate of 76.5%. Based on the preliminary survey, the reliability and validity test results of the initial scale were modified to a certain extent and the final formal survey questionnaire contained 37 items.

The formal research first determined the general distribution of the research objects through stratified sampling so that the sample distribution was reasonable in terms of gender, age, living area, monthly income, living expenditure, and position level. This ensured that the scientific distribution of the research samples was acceptable. The formal questionnaire was distributed from 20 April 2021, to 8 February 2022, and was distributed through an online survey. A questionnaire link and QR code were generated using the domestic professional questionnaire survey platform “wjx.cn”, and the questionnaire link or QR code was sent through WeChat, Weibo, and other social network platforms. At the same time, the questionnaires were distributed through internships and parents’ work groups so that residents from all walks of life would be covered. During the questionnaire survey, the respondents were first informed about the purpose of the study. The results of the survey were only used for scientific research and the respondents’ personal information will not be disclosed. The importance of truthfully and carefully completing the questionnaire was emphasized. A total of 578 questionnaires were collected, of which 503 were valid, with an effective recovery rate of 87.02%. The questionnaires were sorted, and the demographic characteristics of the survey samples are shown in Table 1.

## 4. Empirical Analysis and Results

### 4.1. Reliability and Validity Testing of the Scale

#### 4.1.1. Reliability Analysis of Each Variable

The reliability analysis results showed that the Cronbach’s coefficients of the scale for the residents’ pro-environmental behavior and its influencing factors were all above 0.77, and most of them were above 0.85, which is within the acceptable range. Therefore, the scale has a good reliability. See Table 2 for reliability test index.

#### 4.1.2. Validity Analysis of Each Variable

The validity test was divided into content validity and structural validity. In terms of content validity, the scale in this study was modified based on the literature review and existing mature scales and passed the pre-investigation test. Therefore, the scale in this study has a good content validity. From the perspective of structural validity, this study first investigated the Kaiser–Meyer–Olkin measure of sampling fitness (KMO) and the “Bartlett test of sphericity” (see Table 3). The results showed that the KMO value for each scale was above 0.9, and the significance level of the Bartlett test of sphericity was 0.000, indicating that the effectiveness of each scale had passed the preliminary test and was suitable for factor analysis.

The confirmatory factor analysis of the formal survey was mainly performed using AMOS23.0 software (IBM, Armonk, NY, USA). In accordance with the previous model architecture, the SEM model was drawn and then the sample data obtained from the formal investigation were substituted into the model to test it further. We used the full CFA model, and the factor loading and questionnaire items are shown in Figure 2.

After relevant adjustments to the model, the goodness of fit index of the final output was obtained, as shown in Table 4. It is evident that the fitting indexes for the psychological cluster and the structure of the pro-environmental behavior shown by the residents reached an acceptable level. The standardized load values of the measurement items of each variable are all above 0.75, and the average extraction variation (AVE) of each variable in the psychological clusters and pro-environmental behaviors is also above 0.5, which shows a good convergent validity. In conclusion, the reliability and validity of each variable structure and scale were acceptable.

### 4.2. Research Results and Hypothesis Testing

#### 4.2.1. Correlation Test

Correlation analyses were conducted separately between the independent variables and the mediating variables, the independent variables and the dependent variables, and the mediating variables and the dependent variables in the research model. Figure 3 shows that the independent variables of the model, namely, group emotion and moral belief, are significantly positively correlated with the mediating variables, psychological clusters, and their dimensions. Each independent variable has a significant positive correlation with the pro-environmental behavior shown by the residents and its dimensions. The mediating variable, namely, the psychological cluster and its dimensions, is positively correlated with pro-environmental behavior and its dimensions, and there exists an obvious relationship. In other words, the stronger the psychological cluster, the more obvious the pro-environmental behavior.

#### 4.2.2. Impact Analysis of the Pro-Environmental Behavior Shown by Residents

Table 5 shows the analysis and a summary of the impact of the pro-environmental behaviors. The effect of group emotion on the pro-environmental behavior shown by the residents and its dimensions are all significant. The coefficients for the effect of the group mood on the variable are as follows: pro-environmental behavior = 0.513, basic pro-environmental behavior = 0.420, decision-making pro-environmental behavior = 0.389, interpersonal pro-environmental behavior = 0.454, and citizen pro-environmental behavior = 0.357. Therefore, hypothesis H1 holds. The significance level for moral belief’s effects on pro-environmental behavior and its various dimensions (H2) is below 0.05, and the coefficients for the effect of moral belief on the variable are as follows: pro-environmental behavior = 0.696, Basic pro-environmental Behavior = 0.563, decision-making pro-environmental behavior = 0.520, interpersonal pro-environmental behavior = 0.513, and citizen pro-environmental behavior = 0.533. Therefore, hypothesis H2 holds. The responsibility cluster has a significant effect on pro-environmental behavior. It also has a significant effect on the basic pro-environmental behavior shown by residents, but the effects on decision-making, interpersonal, and citizen pro-environmental behavior are not significant. The coefficients for the effect of the responsibility cluster on the variable are as follows: pro-environmental behavior = 0.235 and basic pro-environmental behavior = 0.246. The emotion cluster and resonance cluster have significant effects on pro-environmental behavior and its dimensions. Therefore, hypothesis H3 partially holds.

#### 4.2.3. Analysis of the Mediating Effects of Psychological Clusters and Their Dimensions

SPSS22.0 (IBM Corp., Armonk, NY, USA) was used to analyze the mediating effect of the mental clusters. The mediating effect was tested by constructing a model containing independent variables (group emotion and moral belief), mediating variables (the psychological cluster), and dependent variables (resident pro-environmental behavior). Pro-environmental behavior had four dimensions: basic pro-environmental behavior, decision-making pro-environmental behavior, interpersonal pro-environmental behavior, and citizen pro-environmental behavior.

Taking the analysis of the mediating effect of psychological cluster on group emotions, moral beliefs, and pro-environmental behavior as an example, this study first constructed a regression equation based on three models, as shown in Table 6, where c is the total effect of X on Y, ab is the mediating effect via the mediating variable M, and c’ is the direct effect. Model 1 represents the regression equation for the total effect, Model 2 represents the regression equation with the psychological cluster as the dependent variable, and Model 3 represents the regression analysis equation with pro-environmental behavior as the dependent variable. The results show that group emotions and moral beliefs have significant, direct impacts on residents’ pro-environmental behavior. However, after the addition of the psychological cluster as a mediating variable, group emotion and moral belief can still significantly affect the residents’ pro-environmental behavior; that is to say, the psychological cluster has a partial mediating effect on the relationship between group emotion, moral belief, and pro-environmental behavior. Table 7 shows the results of the psychological cluster Bootstrap test, which testifies to the above results.

The mediating effects of the psychological cluster on group emotions and moral beliefs, pro-environmental behaviors, and their various dimensions were analyzed, and a summary of the test results is shown in Figure 4. The psychological cluster had partial mediating effects on all the dimensions except for the relationships between group emotion and citizen pro-environmental behavior, moral belief and decision-making pro-environmental behavior, interpersonal pro-environmental behavior, and citizen pro-environmental behavior, where it had a complete mediating effect. Therefore, it can be assumed that both H4 and H5 are true.

## 5. Discussion

The data analysis results of this study show that the overall score for the pro-environmental behavior of the dependent variable is 3.97, which is high, and indicates that most residents actively practice environmental protection behavior in the face of increasingly severe ecological and environmental problems. The average basic pro-environmental behavior score was the highest out of all the pro-environmental behavioral dimensions at 4.12 points, indicating that residents perform better in daily pro-environmental behavior, such as resource conservation, energy, garbage sorting, and other basic environmental behaviors. Compared with the other three dimensions for pro-environmental behaviors, the implementation of basic pro-environmental behaviors costs less and is more convenient.

The correlations between the independent variables and dependent variables show that group emotion has a significant positive effect on the generation of pro-environmental behavior. This is because emotions themselves have a guiding effect on behavior. When individuals are stimulated by environmental problems, they produce corresponding behavioral responses. Harth et al. (2013) found that when individuals perceived that their organization might be engaging in environmentally damaging behavior, their guilt and anger would increase [48]. Guilt will galvanize individuals to repair the environmental damage, while anger may lead individuals to punish the members who cause environmental damage. Ferguson et al. (2010) showed that collective guilt increases individuals’ willingness to conserve energy and pay more environmental taxes [33]. For example, environmental groups tend to induce guilt and highlight that humans should be responsible for climate change, and the media tends to stir up anger over environmental issues, whereas politicians often try to evoke pride in ecological and technological achievements. However, this study does not discuss in depth what kind of pro-environmental behaviors a specific group emotion type would prompt an individual to exhibit. Future studies can build from this aspect to guide the emergence of specific pro-environmental behaviors. Studies have also shown that moral beliefs can significantly predict pro-environmental behavior. People with a high sense of environmental moral belief will more positively influence the environmental behaviors of people around them. However, there have not been many studies on how moral beliefs affect pro-environmental behavior, and previous studies mainly focused on beliefs as moderating variables. At present, many researchers combine group emotions with moral beliefs to explore the influence of moral emotions on pro-environmental behaviors from the perspective of moral emotions. Gerhard and Lisa (2015) proposed that moral anger and a sense of responsibility played an important role in the relationship between beliefs about fairness and pro-environmental behaviors, and anger or dissatisfaction with environmental protection would affect individuals’ willingness to exhibit pro-environmental behaviors [49]. These factors provide possible directions for future research.

This study on the psychological cluster and pro-environmental behavior found that the psychological cluster and its various dimensions can significantly predict basic pro-environmental behaviors. Based on social identity theory, when faced with relatively basic and conveniently solvable environmental problems, individuals psychologically change into collective actors, resulting in a “psychological cluster”. At this time, individual behaviors are easily affected by the group. The intermediary effect analysis showed that group emotion and moral belief directly influenced the environmental behavior of residents and that the psychological cluster, as an intervening variable, significantly affected the group emotional and moral beliefs shown by the residents toward environmental behavior. The psychological cluster effect on the group emotion and moral faith shown by the residents had a partial intermediary effect on environmental behavior. Individuals with the same moral beliefs will have a similar sense of belonging and identity (Liu, 2020; Kyei-Poku, 2020), leading to cluster psychology and similar behaviors. The psychological cluster also had complete mediating effects on the relationship between group emotion and citizen pro-environmental behavior, moral belief and decision-making pro-environmental behavior, interpersonal pro-environmental behavior, and citizen pro-environmental behavior. This result is based on the psychological cluster and is expected to provide a new research object that could influence pro-environmental behavior.

This study explored the influencing mechanism of pro-environmental behaviors from the perspectives of individual psychology and the external environment. The rationality of the sample distribution was ensured as far as possible in this survey, and the sample included education, monthly income, gender, etc. However, there is still room for improvement in the age distribution, and we will supplement it in future research.

## 6. Conclusions

(1) Group emotion and moral belief can significantly predict the pro-environmental behaviors of residents, and moral beliefs have a significant impact on basic pro-environmental behaviors. The coefficients for the effect of group emotion on the variable are as follows: pro-environmental behavior = 0.513, basic pro-environmental behavior = 0.420, decision-making pro-environmental behavior = 0.389, interpersonal pro-environmental behavior = 0.454, and citizen pro-environmental behavior = 0.357.

(2) Each dimension of the psychological cluster has a significant effect on pro-environmental behavior. Among them, the psychological cluster has the most significant effect on the citizen pro-environmental behavior.

(3) The psychological cluster plays a partial mediating role in the relationship between group emotion, moral belief, and the pro-environmental behavior of residents.

## 7. Managerial Implications

As the basic elements of society, residents also bear the social responsibility of environmental protection. The pro-environmental behavior shown by residents plays an important role in solving social environmental problems. The proper guidance of pro-environmental behavior down the right path can fundamentally improve the status quo of environmental pollution and further achieve ecological balance and sustainable development. Based on the research model and the conclusions of this study, a systematic strategy to guide the emergence of pro-environmental behaviors is proposed from the perspectives of group emotions, moral beliefs, and psychological clusters.

(1) Make rational use of group emotions. This conclusion shows that group emotion has a direct or indirect positive effect on pro-environmental behavior. As members of an organizing group, residents are influenced by other members of the group when deciding whether to take certain actions. Therefore, it is necessary to pay full attention to the example and driving role of the pro-environmental residents in the organization, and to create a good environmental protection atmosphere, such as advocating the recycling of office supplies composed of environmental materials and encouraging reductions in disposable tableware. In order to realize the emergence of pro-environmental behaviors, some pressure can be exerted on the groups by publicizing the necessity of environmental protection. It is worth noting that in the process of publicizing the necessity of environmental protection, attention should be paid to the transmission of positive emotions, and the idea of “environmental protection is useless” should be avoided directly or indirectly because it may dampen the enthusiasm generated by pro-environmental behavior.

(2) Establish environmental moral beliefs. The data show that moral beliefs have a positive effect on the pro-environmental behavior shown by residents. Therefore, the state and society should fully guide the residents to establish correct environmental moral beliefs. Moral education should follow internal laws, focus on cultivating the inner belief of residents, give play to individual subjective initiatives, and realize the unity of knowledge and action. Society and enterprises should use certain methods to guide the contradictions and struggles within the thoughts of residents and trigger their independent thinking. In practice, society and enterprises can exert a subtle influence on residents by filming propaganda videos and setting typical examples. Residents should pay attention to the sublimation of moral understanding, internalize moral knowledge into moral belief, cultivate self-education ability, evaluate their own ideology and morality from the perspective of their formed moral belief, put forward self-requirements, and strengthen and improve their inner moral beliefs.

(3) Stimulate the responsibility cluster, promote the emotion cluster, and guide the resonance cluster. The data analysis showed that the psychological cluster and its three subdivided dimensions can positively promote the occurrence of pro-environmental behavior. Therefore, in management practice, organizations should first pay attention to the stimulation of the responsibility psychological clusters and promote the occurrence of pro-environmental behaviors of stimulating individual civic consciousness, the sense of responsibility, and the sense of ownership. Secondly, in order to promote the emotion clusters, society should enhance the love for environmental protection through publicity and the promotion of environmental protection groups. Finally, it is necessary to guide resonance clusters correctly and promote pro-environmental behavior through the unification of personal interests and social and environmental interests.

(4) Cultivate basic and decision-making pro-environmental behaviors and stimulate interpersonal and citizen pro-environmental behaviors. Society should cultivate and encourage strategies based on the basic and decision-making pro-environmental behaviors shown by residents. Specific measures should focus on the cultivation of environmental protection concepts, cultivate small energy-saving and environmental protection habits, and provide certain positive incentives to departments or offices that insist on achieving a certain level of environmental performance. Society needs to adopt incentives and inducement strategies that promote interpersonal and citizen pro-environmental behaviors of residents. Pro-environmental atmospheres need to be created within organizations, and the civic awareness and social responsibility of citizens needs to be encouraged by setting examples of environmental protection and by suggesting incentive mechanisms for environmental protection. Finally, adopting pro-environmental behavior needs to become part of the personal habits and value judgments shown by residents.

## Figures and Tables

**Figure 1 ijerph-19-11190-f001:**
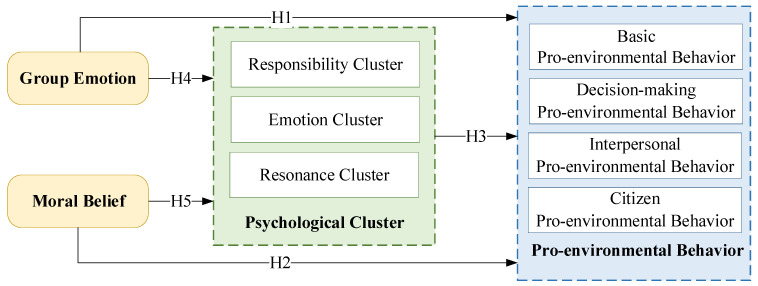
Research model for pro-environmental behavior shown by residents.

**Figure 2 ijerph-19-11190-f002:**
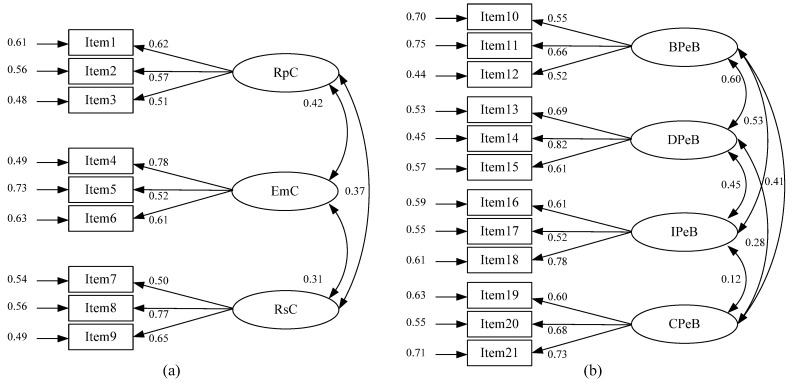
CFA model: (**a**) Psychological cluster; (**b**) Pro-environmental behavior. Note: RpC—Responsibility Cluster, EmC—Emotion cluster, RsC—Resonance cluster, BPeB—Basic Pro-environmental Behavior, DPeB—Decision-making Pro-environmental Behavior, IPeB—Interpersonal Pro-environmental Behavior, and CPeB—Citizen Pro-environmental Behavior.

**Figure 3 ijerph-19-11190-f003:**
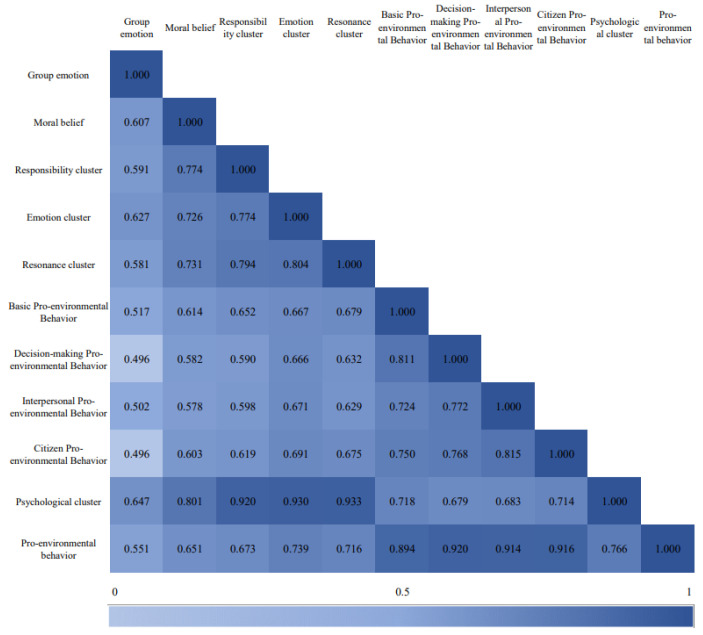
Correlation analysis.

**Figure 4 ijerph-19-11190-f004:**
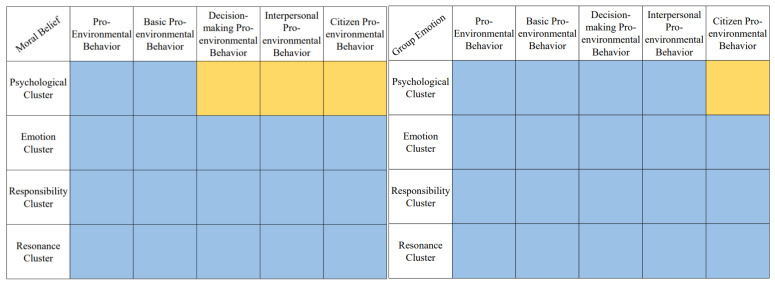
Summary of the psychological cluster’s mediating effect of each dimension. Note: Yellow indicates a complete mediating effect and blue indicates a partial mediating effect.

**Table 1 ijerph-19-11190-t001:** Demographic characteristics of the sample.

Variable		Frequency	Variable		Frequency
Gender	Male	39.4%	Monthly income	≤2000	16.7%
	Female	60.6%		2001–4000	19.3%
Age	≤17	0.1%		4001–6000	19.5%
	18–25	53.7%		6000–8000	18.3%
	26–30	3.8%		8001–10000	15.7%
	31–35	4.0%		10,001–30,000	10.7%
	36–40	5.0%		≥30,000	2.2%
	41–45	7.0%	Record of formal schooling	Junior high school and below	6.2%
	46–50	16.9%		High or technical secondary school	14.5%
	51–60	7.8%		College degree	15.9%
	≥61	1.0%		Full-time undergraduate degree	60.2%
				Master’s degree or above	3.2%

**Table 2 ijerph-19-11190-t002:** Reliability test index.

Variable		Number of Items	Cronbach’s α
Group emotion		3	0.836
Moral belief		3	0.772
Psychological cluster		9	0.941
	Responsibility cluster	3	0.870
	Emotion cluster	3	0.857
	Resonance cluster	3	0.873
Pro-environmental behavior		12	0.953
	Basic Pro-environmental Behavior	3	0.822
	Decision-making Pro-environmental Behavior	3	0.905
	Interpersonal Pro-environmental Behavior	3	0.900
	Citizen Pro-environmental Behavior	3	0.843

**Table 3 ijerph-19-11190-t003:** Bartlett and KMO test results for the formal survey scales.

Scale	Sampling the Kaiser–Meyer–Olkin Measure of Adequacy	Bartlett’s Test for Sphericity		
		The Approximate Chi-Square	df	Sig.
Psychological cluster	0.930	3686.630	36	0.000
Pro-environmental behavior	0.937	5322.092	66	0.000

**Table 4 ijerph-19-11190-t004:** Factor analysis results for the psychological cluster and resident pro-environmental behavior formal survey scales.

	Absolute Fitting Exponent					Incremental Fitting Index		
	χ^2^	χ^2^/df	GFI	RMR	RMSEA	NFI	TLI	CFI
Psychological cluster	200.374	8.349	0.912	0.043	0.121	0.946	0.928	0.952
pro-environmental behavior	476.119	9.919	0.852	0.064	0.133	0.912	0.889	0.920

**Table 5 ijerph-19-11190-t005:** Impact analysis of the pro-environmental behavior shown by residents.

The Dependent Variable	Pro-Environmental Behavior	Basic Pro-Environmental Behavior	Decision-Making Pro-Environmental Behavior	Interpersonal Pro-Environmental Behavior	Citizen Pro-Environmental Behavior
**Linear relation and goodness-of-fit test**					
**F**	215.041	173.701	147.461	147.196	160.015
**Sig.**	0.000	0.000	0.000	0.000	0.000
**R^2^**	0.462	0.410	0.371	0.371	0.390
**Regression coefficient test**					
**t_1_**	5.970	5.275	5.053	5.359	4.664
**Sig.1**	0.000	0.000	0.000	0.000	0.000
**t_2_**	12.160	11.005	9.977	9.701	10.895
**Sig.2**	0.000	0.000	0.000	0.000	0.000
Predictive variables: (constant), group mood (1), moral belief (2)					
**Linear relationships and goodness-of-fit test**					
**F**	243.002	177.623	148.689	151.662	180.164
**Sig.**	0.000	0.000	0.000	0.000	0.000
**R^2^**	0.594	0.516	0.472	0.476	0.520
**Regression coefficient test**					
**t_1_**	2.581	3.768	1.520	1.901	1.504
**Sig.1**	0.010	0.000	0.129	0.058	0.133
**t_2_**	7.947	4.630	7.005	7.199	6.858
**Sig.2**	0.000	0.000	0.000	0.000	0.000
**t_3_**	5.213	5.155	3.727	3.302	5.101
**Sig.3**	0.000	0.000	0.000	0.001	0.000
Predictive variables: (constant), responsibility cluster (1), emotion cluster (2), resonance cluster (3)					

**Table 6 ijerph-19-11190-t006:** Regression analysis of group emotion, moral belief, psychological cluster, and pro-environmental behavior.

	Model 1			Model 2			Model 3		
	B	SE	t	B	SE	t	B	SE	t
Constant term	1.918	0.142	13.490 ***	2.222	0.113	19.634 ***	0.123	0.145	0.848
Group emotion	0.513	0.035	14.773 ***	0.525	0.028	19.004 ***	0.089	0.035	2.539 *
Psychological cluster							0.808	0.043	18.772 ***
R^2^	0.303			0.419			0.591		
F	218.242 ***			361.165 ***			361.859 ***		
Constant term	1.031	0.155	6.629 ***	1.173	0.107	10.971 ***	0.115	0.146	0.785
Moral belief	0.696	0.036	19.208 ***	0.746	0.025	29.952 ***	0.113	0.051	2.214 *
Psychological cluster							0.781	0.055	14.234 ***
R^2^	0.424			0.642			0.590		
F	368.934 ***			897.130 ***			359.994 ***		

Note: * means *p* < 0.05, and *** means *p* < 0.001.

**Table 7 ijerph-19-11190-t007:** Test results for the mediating role played by psychological cluster between independent variables and pro-environmental behavior.

Item	c The Total Effect	a	b	a × b The Mediation Effect	a × b (95% BootCI)	c’ Direct Effect	Inspection Conclusion
Group emotion ≥ psychological cluster ≥ pro-environmental behavior	0.513 ***	0.525 ***	0.808 ***	0.424	0.383~0.532	0.089 *	Partial mediating effect
Moral beliefs ≥ psychological cluster ≥ pro-environmental behavior	0.696 ***	0.746 ***	0.781 ***	0.583	0.444~0.631	0.113	Partial mediating effect

Note: * means *p* < 0.05, and *** means *p* < 0.001.

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
