# Peer review of "Effects of Group Emotion and Moral Belief on Pro-Environmental Behavior: The Mediating Role of Psychological Clustering"

_ijerph, 2022, doi:10.3390/ijerph191811190_

Round 1
Reviewer 1 Report (Previous Reviewer 2)
The paper properly analyzes important issues for the influence of group emotions and moral beliefs on pro-environmental behavior by residents and introduces the concept of the “psychological cluster” to explore the mediating effect of “psychological cluster” on group emotions, moral beliefs, pro-environmental behavior by residents.
The article is well structured. However, It needs to revise descriptive statistics, Empirical analysis and results.
The details are as follows.
In the Literature Review and the methods are adequately described.
The age group of the survey subjects is excessively large with 53.7% of those aged 18-25 years. It is necessary to mention whether this bias does not affect the objectivity of the study.
It is necessary to check and explain discriminant validity of the constructs (highest squared correlation of the variable vs. AVE).
In discussion, the authors should comment on the results of the hypothesis obtained by referring to previous studies, thus confirming what other research has shown.
Author Response
Many thanks to the reviewer for your positive recognition and full affirmation of my manuscript. We feel much honored that the reviewer made positive comments on the significance of topic selection. The comments are valuable and very helpful for revising and improving the quality of our paper. We have studied comments carefully and have made revisions thoroughly for our manuscript in accordance with reviewers' suggestions. In particular, we have revised descriptive statistics, empirical analysis and results.
Point 1: In the Literature Review and the methods are adequately described.
Response 1:
Thanks for your kind comment.
Point 2: The age group of the survey subjects is excessively large with 53.7% of those aged 18-25 years. It is necessary to mention whether this bias does not affect the objectivity of the study.
Response 2:
Thanks for your recommendation. This study explores the influence mechanism of pro-environmental behaviors from the perspectives of individuals psychology and external environment. The rationality of the sample distribution was ensured as soon as possible during the survey, including education, monthly income, gender, etc. However, there is still room for improvement in age distribution, and we will supplement it in future research. We have addressed this study limitation in the article.
Point 3: It is necessary to check and explain discriminant validity of the constructs (highest squared correlation of the variable vs. AVE).
Response 3:
Thank you for your valuable comments. We have checked the discriminant validity. The standardized load values of the measurement items of each variable are all greater than 0.75, and the average extraction variation (AVE) of each variable in psychological clusters and pro-environmental behavior is greater than 0.5, showing good convergent validity.
Point 4: In discussion, the authors should comment on the results of the hypothesis obtained by referring to previous studies, thus confirming what other research has shown.
Response 4:
Thanks for your suggestion, we have commented on the results of the hypothesis obtained by referring to previous studies.
This manuscript is a resubmission of an earlier submission. The following is a list of the peer review reports and author responses from that submission.
Round 1
Reviewer 1 Report
Dear authors,
Well conducted research.
Overall, this is a well written research study, however, I have noted few issues which need to be addressed by authors before publishing this manuscript. I will be more than happy review the second revision of this manuscript. Following are my comments for authors to improve the manuscript:
1. At the end of line 114 year of publication of Chen et al. is missing.
2. Heading 6. (Research suggestions) are actually managerial implications. You need to change this heading.
3. Subheadings from 2.2 to 2.5 need to be rephrased. Do not sounds well.
4. You need to add the full CFA model showing the factor loading along with the questionnaire items.
5. You also need to show the SEM model showing the fit indices. You need to make it clear whether you tested full or partial mediation model. Also show the difference in fit indices of full and partial mediation model. Lastly, you need to show which one (full or partial) mediation model was selected for analysis.
6. In regression analysis, instead of saying "The significance level for moral belief effects on pro-environmental behavior and its various dimensions (H2) was less than 0.05", you need to write the Coefficient (Beta) value for the relationship between the two variables of hypothesis. Do the same for all hypotheses.
7. I will recommend you to explain more about the mediation analysis. Results are not described clearly.
8. Professional proof read is needed for this manuscript to remove the potential language/grammar mistakes before publishing.
9. Lastly, add one paragraph conclusion at the end.
Cheers
Reviewer
Author Response
Many thanks to the reviewer for your positive recognition and full affirmation of my manuscript. We feel much honored that the reviewer made positive comments on the significance of topic selection. The comments are valuable and very helpful for revising and improving the quality of our paper. We have studied comments carefully and have made revisions thoroughly for our manuscript in accordance with reviewers' suggestions.
Point 1: At the end of line 114 year of publication of Chen et al. is missing.
Response 1:
Thanks for your suggestions. We have added the year of publication of Chen et al.
Point 2: Heading 6. (Research suggestions) are actually managerial implications. You need to change this heading.
Response 2:
Thank you for your valuable comments. We have changed “Research Suggestions” to “Managerial Implications”.
Point 3: Subheadings from 2.2 to 2.5 need to be rephrased. Do not sounds well.
Response 3:
Thank you for your valuable comments. We have rephrased the subheadings from 2.2 to 2.5.
Point 4: You need to add the full CFA model showing the factor loading along with the questionnaire items.
Response 4:
Thanks for your suggestions. We have added the full CFA model in our paper.
Point 5: You also need to show the SEM model showing the fit indices. You need to make it clear whether you tested full or partial mediation model. Also show the difference in fit indices of full and partial mediation model. Lastly, you need to show which one (full or partial) mediation model was selected for analysis.
Response 5:
Thanks for your suggestions. The mediating effects between variables in this study were analyzed using SPSS22.0 with Process3.3 software, not by constructing a SEM model (using AMOS software). Our reference for this method is "Trivedi J, Sama R. Determinants of consumer loyalty towards celebrity-owned restaurants: The mediating role of brand love. Journal of Consumer Behaviour. 2021, 20:748-761", with a regression analysis of direct and indirect effects and a Bootstrap test to demonstrate the relevant mediating effect (see section 4.2.3). One way to determine whether the effect is partially mediated or fully mediated is whether the independent and mediating variables have a significant effect on the dependent variable. If both the independent and mediating variables have a significant effect on the dependent variable, it is partially mediated; if the independent variable can significantly affect the mediating variable and only the mediating variable has a significant effect on the dependent variable, it is fully mediated.
Point 6: In regression analysis, instead of saying "The significance level for moral belief effects on pro-environmental behavior and its various dimensions (H2) was less than 0.05", you need to write the Coefficient (Beta) value for the relationship between the two variables of hypothesis. Do the same for all hypotheses.
Response 6: Thank you for your valuable comments. We have written the Coefficient (Beta) value for the relationship between the two variables of hypothesis. The details are as follows:
“The effect of group emotion on the pro-environmental behavior shown by residents and its dimensions are all significant. The coefficients for the effect of group mood on the variable are as follows: pro-environmental behavior = 0.513, Basic pro-environmental Behavior = 0.420, Decision-making Pro-environmental Behavior = 0.389, Interpersonal Pro-environmental Behavior =0.454, and Citizen Pro-environmental Behavior = 0.357. Therefore, hypothesis H1 holds. The significance level for moral belief effects on pro-environmental behavior and its various dimensions (H2) is below 0.05, and the coefficients for the effect of moral belief on the variable are as follows: pro-environmental behavior = 0.696, Basic pro-environmental Behavior = 0.563, Decision-making Pro-environmental Behavior = 0.520, Interpersonal Pro-environmental Behavior =0.513, and Citizen Pro-environmental Behavior = 0.533. Therefore, hypothesis H2 holds. Responsibility cluster has a significant effect on pro-environmental behavior. They also have a significant effect on the basic pro-environmental behavior shown by residents, but the effects on decision-making, interpersonal, and citizen pro-environmental behavior are not significant. the coefficients for the effect of responsibility cluster on the variable are as follows: pro-environmental behavior = 0.235, Basic pro-environmental Behavior = 0.246. Emotion cluster and resonance cluster have significant effects on pro-environmental behavior and its dimensions. Therefore, hypothesis H3 partially holds.”
Point 7: I will recommend you to explain more about the mediation analysis. Results are not described clearly.
Response 7:
Thank you for your valuable comments. According to your suggestion, we have explained more about the mediation analysis. The details are as follows:
“c is the total effect of X on Y, ab is the mediating effect via the mediating variable M, and c' is the direct effect. Model 1 represents the regression equation for total effect, Model 2 represents the regression equation with psychological cluster as the dependent variable, and Model 3 represents the regression analysis equation with pro-environmental behavior as the dependent variable. The results show that group emotions and moral beliefs have significant, direct impacts on residents’ pro-environmental behavior. However, after the addition of psychological cluster as a mediating variable, group emotion and moral belief can still significantly affect residents’ pro-environmental behavior, that is to say, psychological cluster has a partial mediating effect on the relationship between group emotion, moral belief, and pro-environmental behavior.”
Point 8 :Professional proof read is needed for this manuscript to remove the potential language/grammar mistakes before publishing.
Response 8:
Thanks for your valuable comments. We are so sorry for vague expression of this sentence, and the language barrier confused the reviewer. We have consulted a professional language editing service to check the English.
Point 9: Lastly, add one paragraph conclusion at the end.
Response 9:
Thank you for your valuable comments. We have added one paragraph conclusion at the end. The details are as follows:
“6.1 Conclusions
(1) Group emotion and moral belief can significantly predict pro-environmental behaviors by residents, and moral belief have a significant impact on basic pro-environmental behaviors. The coefficients for the effect of group emotion on the variable are as follows: pro-environmental Behavior = 0.513, Basic pro-environmental Behavior = 0.420, Decision-making Pro-environmental Behavior = 0.389, Interpersonal Pro-environmental Behavior =0.454, and Citizen Pro-environmental Behavior = 0.357.
(2) Each dimension of the psychological cluster has a significant effect on pro-environmental behavior. Among them, the psychological cluster has the most significant effect on the citizen pro-environmental behavior.
(3) Psychological cluster plays a partial mediating role in the relationship between group emotion, moral belief, and pro-environmental behavior by residents.”
Reviewer 2 Report
The paper properly analyzes important issues for the influence of group emotions and moral beliefs on pro-environmental behavior by residents and introduces the concept of the “psychological cluster” to explore the mediating effect of “psychological cluster” on group emotions, moral beliefs, pro-environmental behavior by residents.
The article is well structured. However, It needs to revise research model, descriptive statistics, Empirical analysis and results.
The details are as follows.
In the Literature Review: Researchers should include H1~H5 in Figure 1.
The methods are adequately described.
The age group of the survey subjects is excessively large with 53.7% of those aged 18-25 years. It is necessary to mention whether this bias does not affect the objectivity of the study.
It is necessary to check and explain discriminant validity of the constructs (highest squared correlation of the variable vs. AVE).
In discussion, the authors should comment on the results of the hypothesis obtained by referring to previous studies, thus confirming what other research has shown.
Author Response
Many thanks to the reviewer for your positive recognition and full affirmation of my manuscript. We feel much honored that the reviewer made positive comments on the significance of topic selection. The comments are valuable and very helpful for revising and improving the quality of our paper. We have studied comments carefully and have made revisions thoroughly for our manuscript in accordance with reviewers' suggestions.
Point 1: In the Literature Review: Researchers should include H1~H5 in Figure 1.
Response 1:
Thanks for your suggestion, we have presented the hypothesis (H1~H5) in Figure 1.
Point 2: The methods are adequately described.
Response 2:
Thanks for your kind comment.
Point 3: The age group of the survey subjects is excessively large with 53.7% of those aged 18-25 years. It is necessary to mention whether this bias does not affect the objectivity of the study.
Response 3:
Thanks for your recommendation. This study explores the influence mechanism of pro-environmental behaviors from the perspectives of individuals psychology and external environment. The rationality of the sample distribution was ensured as soon as possible during the survey, including education, monthly income, gender, etc. However, there is still room for improvement in age distribution, and we will supplement it in future research. We have addressed this study limitation in the article.
Point 4: It is necessary to check and explain discriminant validity of the constructs (highest squared correlation of the variable vs. AVE).
Response 4:
Thank you for your valuable comments. We have checked the discriminant validity. The standardized load values of the measurement items of each variable are all greater than 0.75, and the average extraction variation (AVE) of each variable in psychological clusters and pro-environmental behavior is greater than 0.5, showing good convergent validity.
Point 5: In discussion, the authors should comment on the results of the hypothesis obtained by referring to previous studies, thus confirming what other research has shown.
Response 5:
Thanks for your suggestion, we have commented on the results of the hypothesis obtained by referring to previous studies.